# Structure of the human epithelial sodium channel by cryo-electron microscopy

**Sigrid Noreng[1], Arpita Bharadwaj[2], Richard Posert[1], Craig Yoshioka[3], Isabelle Baconguis[2]\***

[1]Department of Biochemistry & Molecular Biology, Oregon Health and Science University, Portland, United States; [2]Vollum Institute, Oregon Health and Science University, Portland, United States; [3]Department of Biomedical Engineering, Oregon Health and Science University, Portland, United States

**Abstract** The epithelial sodium channel (ENaC), a member of the ENaC/DEG superfamily, regulates $Na^+$ and water homeostasis. ENaCs assemble as heterotrimeric channels that harbor protease-sensitive domains critical for gating the channel. Here, we present the structure of human ENaC in the uncleaved state determined by single-particle cryo-electron microscopy. The ion channel is composed of a large extracellular domain and a narrow transmembrane domain. The structure reveals that ENaC assembles with a 1:1:1 stoichiometry of α:β:γ subunits arranged in a counter-clockwise manner. The shape of each subunit is reminiscent of a hand with key gating domains of a 'finger' and a 'thumb.' Wedged between these domains is the elusive protease-sensitive inhibitory domain poised to regulate conformational changes of the 'finger' and 'thumb'; thus, the structure provides the first view of the architecture of inhibition of ENaC.
DOI: https://doi.org/10.7554/eLife.39340.001

**\*For correspondence:**
bacongui@ohsu.edu

**Competing interests:** The authors declare that no competing interests exist.

## Introduction

The fine-tuning of $Na^+$ homeostasis is largely mediated by epithelial sodium channels (ENaC) that are related in amino acid sequence to acid-sensing ion channels (ASIC) found in eukaryotes, degenerin channels (DEG) of *Caenorhabditis elegans*, and the FMRF-amide peptide-gated channels (FaNaCh) of mollusk (*Driscoll and Chalfie, 1991*; *Chalfie and Wolinsky, 1990*; *Kellenberger and Schild, 2002*; *Waldmann et al., 1997a*; *Waldmann et al., 1997b*; *Krishtal and Pidoplichko, 1981*; *Chelur et al., 2002*; *Garty and Palmer, 1997*; *Cottrell et al., 1990*; *Lingueglia et al., 1995*). These ion channels belong to the voltage-independent, $Na^+$-selective, and amiloride-sensitive ENaC/DEG superfamily which together perform diverse cellular functions in different organisms. In humans, ENaCs are expressed at the apical surface of epithelial tissues throughout the body, and play critical roles that range from regulation of total-body salt, water, and blood volume, to modulating airway surface liquid clearance in epithelial cells in the lungs (*Büsst, 2013*; *Ismailov et al., 1996*; *Rossier et al., 2015*; *McDonald et al., 1994*). The importance of ENaC in $Na^+$ homeostasis is highlighted by gain-of-function mutations causing severe hypertension, as in Liddle syndrome, or loss-of-function mutations causing the neonatal salt-wasting disorder pseudohypoaldosteronism type 1 (PHA-1) (*Gründer et al., 1997*; *Hansson et al., 1995*; *Shimkets et al., 1994*; *Chang et al., 1996*; *Edelheit et al., 2005*; *Kerem et al., 1999*). More subtle ENaC dysfunction contributes to diseases as diverse as essential hypertension, heart failure, and nephrotic syndrome (*Soundararajan et al., 2010*; *Hamm et al., 2010*; *Zheng et al., 2016*). ENaCs require three different subunits to form a functional channel, α, β, and γ (*Canessa et al., 1994*). Despite decades of study, the number of subunits in an active channel remains unclear (*Shobair et al., 2016*).

Unique among the ENaC/DEG channels, ENaCs are activated by proteolysis of peptidyl tracts embedded in the extracellular domain (ECD), which releases inhibitory peptides. The cleavage event

**eLife digest** The bodies of humans and other animals contain many different fluids that play vital roles in the body, such as blood, saliva and the fluids that surround cells in organs. These fluids all contain particles called ions, which can affect the flow of water into and out of cells and alter the activity of proteins. Therefore, in order to survive, an animal must tightly regulate the levels of ions in its body.

Epithelial cells line the surface of organs, and the inside of the digestive system and other cavities in the human body. A channel known as ENaC is found on the surface of epithelial cells and controls the volume of the fluid surrounding cells, blood pressure and the volume of liquid in the airways. This channel spans the membrane surrounding each epithelial cell and allows sodium ions to pass into the cell. To promote the opening of the channel, enzymes remove portions of the ENaC called extracellular domain, which sits on the outside surface of an epithelial cell. Three components (or 'subunits') called alpha, beta and gamma are needed to form an ENaC, but it is not clear how they fit together to form a single working unit.

Noreng et al. used a technique called cryo-electron microscopy to study the three-dimensional structure of the human ENaC. This revealed that a single channel contains one alpha, one beta and one gamma subunit, which sit next to each other to form a narrow tube through the membrane and a large extracellular domain. When viewed from the outside of the cell the subunits form a narrow ring in a counter-clockwise manner.

Further analysis of the structure suggested that when enzymes remove pieces of the extracellular domain of ENaC, it becomes easier for the rest of the channel to adopt a shape that allows sodium ions to move through the pore. A next step will be to study the three-dimensional structure of ENaC when it takes on different shapes to better understand how it works.

DOI: https://doi.org/10.7554/eLife.39340.002

increases channel opening probability, $P_o$ (*Orce et al., 1980*; *Vallet et al., 1997*; *Vallet et al., 2002*; *Vuagniaux et al., 2002*; *Hughey et al., 2004*; *Hughey et al., 2003*; *Caldwell et al., 2004*; *Passero et al., 2010*; *Kleyman et al., 2009*). Amino acid sequence alignments and biochemical analyses in the ECD have so far revealed that only the β subunit lacks the characteristic motifs for protease recognition. ENaCs are widely known as substrates of serine proteases like furin, and a growing list of proteases that recognize sites in ENaC suggests a multifaceted regulation of channel function (*Rossier and Stutts, 2009*). Indeed, the complexities of ENaC function involving the requisite heteromeric subunit assembly and asymmetric subunit modification via differential proteolytic processing are critical to ion channel gating. Thus, to define subunit arrangement and stoichiometry, and elucidate the molecular architecture of ENaC inhibition, we determined the structure of ENaC in the uncleaved state by single-particle cryo-electron microscopy (cryo-EM).

## Results

### Design and expression of ΔENaC

We first assessed the expression of full-length (FL) ENaC by small-scale expression in adherent HEK293S GnTI⁻ cells and fluorescence-detection size-exclusion chromatography (FSEC) (*Kawate and Gouaux, 2006*). We found a low, wide peak, indicating a poorly expressing polydisperse population unsuitable for cryo-EM (*Figure 1a*). We thus screened a number of deletions and mutations in each ENaC subunit, harnessing information derived from previous biochemical and functional experiments gauging the propensity for heterotrimeric formation of ENaC and its susceptibility to proteolytic processing (*Canessa et al., 1994*; *Orce et al., 1980*; *Vallet et al., 1997*; *Vallet et al., 2002*; *Vuagniaux et al., 2002*; *Hughey et al., 2004*; *Hughey et al., 2003*; *Caldwell et al., 2004*; *Passero et al., 2010*), before arriving at the construct referred to here as ΔENaC (*Figure 1a–c*, *Figure 1—figure supplement 1*, *Figure 1—figure supplement 2*).

ΔENaC is composed of αβγ subunits truncated at the N- and C-termini (*Figure 1b,c*). Additionally, the Δα and Δγ subunits possess mutations in the identified furin and prostasin sites which prevent subunit cleavage and channel activation (*Hughey et al., 2003*; *Bruns et al., 2007*). For protein

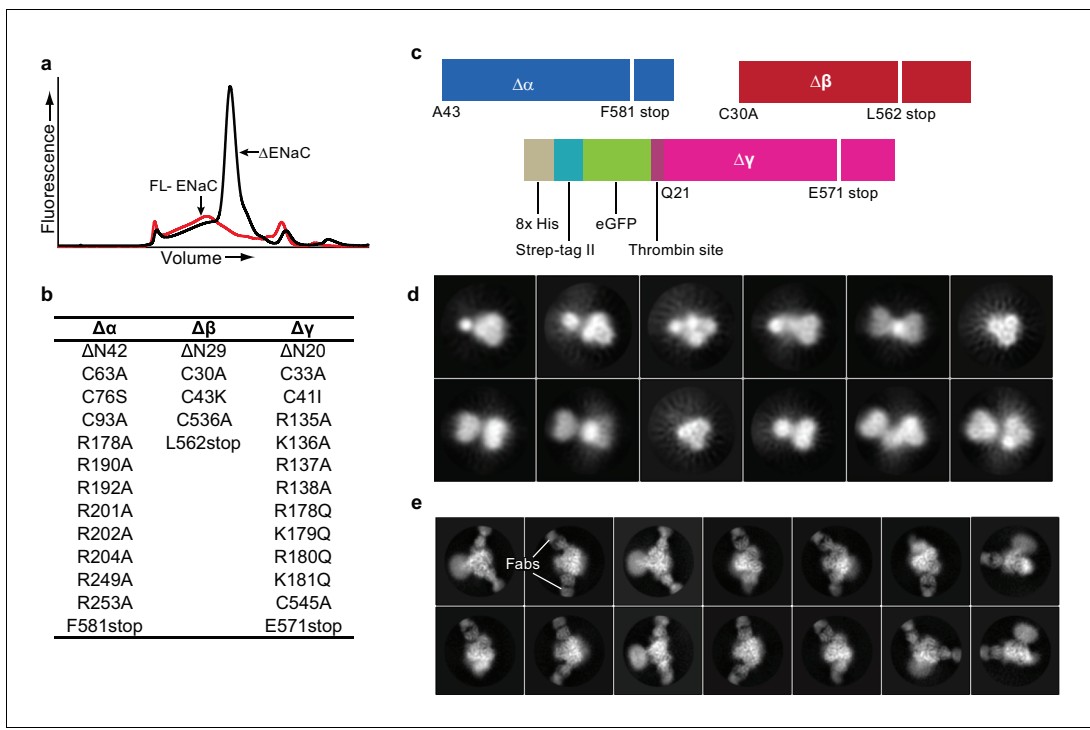

**Figure 1.** Creation and analysis of ΔENaC. (**a**) Representative FSEC traces of full-lenth ENaC (FL-ENaC, red) and ΔENaC (black). (**b**) Summary of mutations in ΔENaC. (**c**) Summary of ΔENaC constructs. (**d**) Representative 2D class averages of ΔENaC show that pseudosymmetry inherent in ENaC hampers particle alignment. (**e**) Representative 2D class averages of ΔENaC-7B1/10D4 complex showing increased detail due to alignment aid from Fabs.

DOI: https://doi.org/10.7554/eLife.39340.003

The following figure supplements are available for figure 1:

**Figure supplement 1.** Sequence alignment of ENaC with other members of the ENaC/DEG superfamily (human ENaCα residues 1-387).

DOI: https://doi.org/10.7554/eLife.39340.004

**Figure supplement 2.** Sequence alignment of ENaC with other members of the ENaC/DEG superfamily (human ENaCα residues 388-669)

DOI: https://doi.org/10.7554/eLife.39340.005

**Figure supplement 3.** Fab generation.

DOI: https://doi.org/10.7554/eLife.39340.006

**Figure supplement 4.** Fab binding properties.

DOI: https://doi.org/10.7554/eLife.39340.007

purification, neither Δα nor Δβ were modified with affinity tags because there is strong evidence that the α subunit can readily form functional homomeric channels, and the termini of Δβ are sensitive to perturbations (*Canessa et al., 1993*). As a result, Δγ contains both GFP and a Strep-II tag at the N-terminus (*Figure 1c*), minimizing contamination by homomeric Δα channels during purification. This construct provided a homogeneous and highly-expressing population. However, the inherent pseudosymmetry from common secondary and tertiary structures between the α, β, and γ subunits of human ENaC hindered particle alignment (*Figure 1d*).

To evaluate biochemical integrity and to facilitate cryo-EM three-dimensional reconstruction of ΔENaC, we generated subunit-specific monoclonal antibodies (mAbs) that bind to three-dimensional epitopes in ΔENaC and FL-ENaC. For immunization, we exploited the high-expressing chicken ASIC (cASIC) by adding the first 22 N-terminal amino acids of cASIC to Δβ, which tolerated the fusion. This construct is referred to hereafter as Δβ_ASIC. Together, Δα, Δβ_ASIC, and Δγ comprise ΔENaC_ASIC (*Figure 1—figure supplement 3a,b*) (*Jasti et al., 2007*). Two fragment-antigen binding domains (Fabs) were isolated that recognize different epitopes (7B1 and 10D4). While these antibodies were raised against ΔENaC_ASIC (*Figure 1—figure supplement 3a,b*), both Fabs bind to both ΔENaC

expressed in HEK 293S GnTI⁻ and FL-ENaC expressed in HEK293T/17, which indicates that ΔENaC is properly folded and that the Fabs do not bind to the ASIC segment (*Figure 1—figure supplement 3c,d*; *Figure 1—figure supplement 4a,b*). Inclusion of 7B1 and 10D4 allowed for proper alignment of the particles (*Figure 1e*). Moreover, maps of the particles with only 10D4 (monoFab) compared to those with both 10D4 and 7B1 (diFab) show that each Fab recognizes only one subunit (*Figure 1—figure supplement 4c–f*). We monitored and compared grid conditions and the resulting data quality (including ice thickness, sample quality, particle distribution, and orientation) between the mono-Fab and the diFab complexes of ENaC and discovered that the diFab complex was a more promising complex for cryo-EM analysis.

## Functional characterization of ΔENaC

We investigated ΔENaC function by two-electrode voltage-clamp electrophysiology (TEVC) and whole-cell patch clamp electrophysiology in oocytes and GnTI⁻ HEK cells, respectively (*Figure 2*, *Figure 2—figure supplements 1* and *2a and b*). Unlike FL-ENaC (*Figure 2a*, *Figure 2—figure supplement 2a*), ΔENaC does not exhibit amiloride-sensitive currents in oocytes and HEK cells, and the only Na⁺-specific currents resemble those from uninjected oocytes (*Figure 2—figure supplement 1a,b*). Similarly, oocytes expressing ENaC channels with restored protease sites in the Δ subunits (Δα* and Δγ*) to form Δ*ENaC did not present amiloride-sensitive currents (*Figure 2—figure supplement 1c*). Because HEK cells are better suited to defining whether ΔENaC traffics to the plasma membrane, we examined surface expression of ΔENaC and FL-ENaC expressed in GnTI⁻ HEK cells using confocal microscopy. To ensure robust expression, we transduced the HEK cells with baculovirus encoding the ΔENaC and FL-ENaC proteins, taking advantage of the N-terminal eGFP in the Δγ subunit and the N-terminal eGFP in all three FL subunits to visualize expression, respectively. Based on eGFP fluorescence, we observed robust expression of both ΔENaC and FL-ENaC (*Figure 2—figure supplement 2c,d*). We employed tetramethylrhodamine (TRITC)-labeled 10D4 mAb, an antibody that binds to the extracellular domain of ENaC, to probe the plasma membrane localization of ENaC channels. Indeed, we observed overlapping signals from both eGFP and TRITC-10D4 mAb in cells expressing FL-ENaC but not in cells expressing ΔENaC. Based on the confocal imaging results, ΔENaC is not trafficked to the plasma membrane, in agreement with the electrophysiology results in HEK 293S GnTI⁻ cells and oocytes (*Figure 2—figure supplement 2*).

We further examined whether disruption of the channel by mutagenesis also caused the absence of ΔENaC current. We tested channels comprising a single Δ or Δ* subunit in complex with the two complementary FL-ENaC subunits. Channels comprising Δα-FLβ-FLγ conduct amiloride-sensitive Na⁺ currents which increase approximately 5-fold upon trypsin treatment (compared with 2.2-fold for FL-ENaC, *Figure 2b*, *Figure 2—figure supplements 2* and *3*, and *Figure 2—source data 1*). Since this trypsin response could be a result of cleavage of FLγ, we also tested channels of Δα*-FLβ-FLγ (*Figure 2c*). These channels show an increase in total current compared to Δα-FLβ-FLγ, and demonstrate a more archetypal ENaC current trace (*Figure 2a*, *Figure 2—figure supplement 3*, *Figure 2—source data 1*). These results, in addition to the cleavage pattern of an anti-α immunoblot (*Figure 2—figure supplement 4*) indicate that Δα adopts a biologically relevant fold, capable of forming active channels with other full-length subunits, and that it is likely cleaved once at its N-terminal furin site (RSRA in Δα) but not the C-terminal furin site (AAAA in Δα, *Figure 1—figure supplement 1*). By restoring the protease sites, as in Δα*, the inhibitory peptide was effectively removed.

The FLα-Δβ-FLγ channels conducted amiloride-sensitive Na⁺ currents with a post-trypsin/pre-trypsin ratio of 1.5 (*Figure 2d*, *Figure 2—figure supplement 3*, *Figure 2—source data 1*), similar to that of FL-ENaC. Moreover, an anti-β immunoblot shows no cleavage of Δβ, as expected (*Figure 2—figure supplement 5*). The FLα-FLβ-Δγ channel also conduct an amiloride-sensitive Na⁺ current with approximately 9.5-fold increase upon trypsin treatment (*Figure 2e*, *Figure 2—figure supplements 2* and *3*, and *Figure 2—source data 1*). Although the Δγ subunit has the canonical furin and prostasin sites mutated (AAAA and QQQQ respectively, *Figure 1—figure supplement 1*), there are other basic residues near the furin and prostasin sites that could be cleaved by trypsin. This hypothesis is further supported by the immunoblot showing significant trypsin digestion in Δγ (*Figure 2—figure supplement 6*) as well as the even higher trypsin activation of FLα-FLβ-Δγ* (approximately 13.3-fold, *Figure 2f* and *Figure 2—figure supplement 3* and *Figure 2—source data 1*). Nevertheless, the results are a promising direction for future studies. Importantly, the combination of TEVC traces of

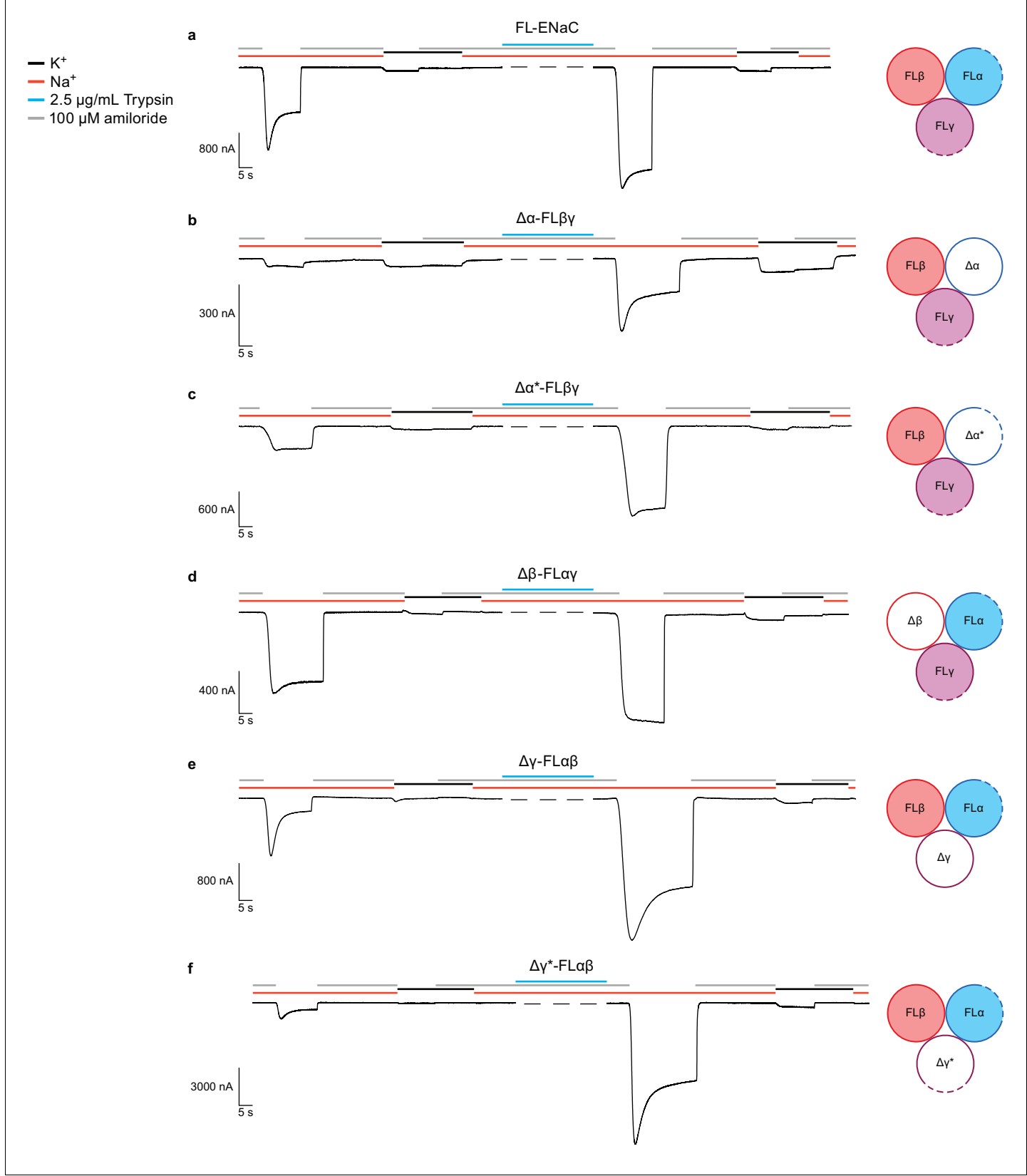

**Figure 2.** Functional characterization of ΔENaC by TEVC. (a) Representative current trace of FL-ENaC shows selectivity of Na⁺ over K⁺, block by amiloride and sensitivity to trypsin treatment (2.5 μg/mL for 5 min) by a 2.22 ± 0.49 fold increase in steady state currents post-trypsin treatment (n = 3).
*Figure 2 continued on next page*

*Figure 2 continued*

The cartoon located on the right side of each current trace represents the combination of subunits injected in the oocytes. Filled circles represent FL-ENaC subunits while open represent the ΔENaC subunits. Dotted lines represent ENaC subunits that contain the intact protease sites. (**b–f**) Representative current traces of Δα-FLβγ (**b**), Δα*-FLβγ (**c**), Δβ-FLαγ (**d**), Δγ-FLαβ (**e**) and Δγ*-FLαβ (**f**) demonstrate that the ΔENaC subunits can form a functional channel with two FL-ENaC subunits that are selective for $Na^+$ over $K^+$ and sensitive to amiloride and trypsin treatment. Currents after trypsin treament increased by 5.15 ± 1.13 (**b**), 4.42 ± 0.61 (**c**), 1.46 ± 0.1 (**d**), 9.52 ± 2.88 (**e**) and 13.26 ± 5.67 (**f**) fold (n = 3 for all combinations).

DOI: https://doi.org/10.7554/eLife.39340.008

The following source data and figure supplements are available for figure 2:

**Source data 1.** Ratio of measured steady state currents pre- and post trypsin treatment.
DOI: https://doi.org/10.7554/eLife.39340.015

**Figure supplement 1.** Currents measured in oocytes injected with ΔENaC resembles currents observed in uninjected oocytes.
DOI: https://doi.org/10.7554/eLife.39340.009

**Figure supplement 2.** FL-ENaC and ΔENaC trafficking in HEK 293S GnTI⁻ cells.
DOI: https://doi.org/10.7554/eLife.39340.010

**Figure supplement 3.** Functional characterization of ΔENaC by TEVC.
DOI: https://doi.org/10.7554/eLife.39340.011

**Figure supplement 4.** Cleavage of ENaC Δα and FL-α by trypsin shows expected banding.
DOI: https://doi.org/10.7554/eLife.39340.012

**Figure supplement 5.** Cleavage of ENaC Δβ and FL-β by trypsin shows expected banding.
DOI: https://doi.org/10.7554/eLife.39340.013

**Figure supplement 6.** Cleavage of ENaC Δγ and FL-γ by trypsin shows expected bands.
DOI: https://doi.org/10.7554/eLife.39340.014

each Δ subunit and the α and γ Δ* counterparts supports ΔENaC representing a biologically relevant channel.

## Cryo-EM analysis of ΔENaC

We solved the structure of ΔENaC diFab complex in n-Dodecyl β-D-maltoside (DDM) by cryoEM (*Figure 3*, *Figure 3—source data 1*). We first carried out cycles of 2D and 3D classifications to remove ice contamination, micelles, and denatured complexes. The remaining particles were subjected to unsupervised ab initio 3D classification and refinement in cryoSPARC (*Punjani et al., 2017*) as well as 3D classification and refinement in cisTEM (*Grant et al., 2018*) to arrive at the cryo-EM potential map with a nominal resolution of 4.2 Å from both programs, based on the gold standard FSC = 0.143 and solvent adjusted FSC = 0.143 criteria, respectively (*Figure 3—figure supplement 1*). Additionally, we conducted a masked refinement excluding the flexible Fc domains of the Fabs and micelle in cisTEM (*Figure 3a*), and obtained a map at 3.9 Å, as determined by the solvent adjusted FSC = 0.143 criterion (*Figure 3b,c*), with local resolution estimates generated by BSoft (*Heymann and Belnap, 2007*) indicating regions of the map with a resolution of 3.7 Å (*Figure 3d*).

## Discussion

### ENaC structural overview

The cryo-EM potential map has three major regions into which the two Fabs and homology models of ΔENaC were manually fitted (*Figure 4*). Alignment of predicted glycosylation sites and aromatic residues to distinct features in the map allowed for the correct assignment of the homology models of the ENaC ECD, generated from the desensitized state of ASIC (PDB: 2QTS, *Figure 4—figure supplements 1–5*, *Figure 4—video 1*). The β subunit is predicted to have 11 glycosylation sites by primary sequence, considerably more than α or γ. Six prominent glycosylation sites were used to assign β (as opposed to the three each in α and γ), whereas a glycosylation on the β9-α4 loop distinguished α from γ (*Figure 1—figure supplement 2*, *Figure 4—figure supplement 1*). Guided by these features and the 10D4 monoFab ΔENaC map, we assigned the identity of 7B1 and 10D4 as binding α and β subunits, respectively (*Figure 4*, *Figure 1—figure supplement 4*).

Forming a trimeric ensemble, the α-β-γ subunits arrange in a counterclockwise manner, as reported by previous studies (*Collier and Snyder, 2011*; *Collier et al., 2014*; *Chen et al., 2011*) (*Figure 4b,d*). The overall domain organization within each subunit of ΔENaC concurs with that of

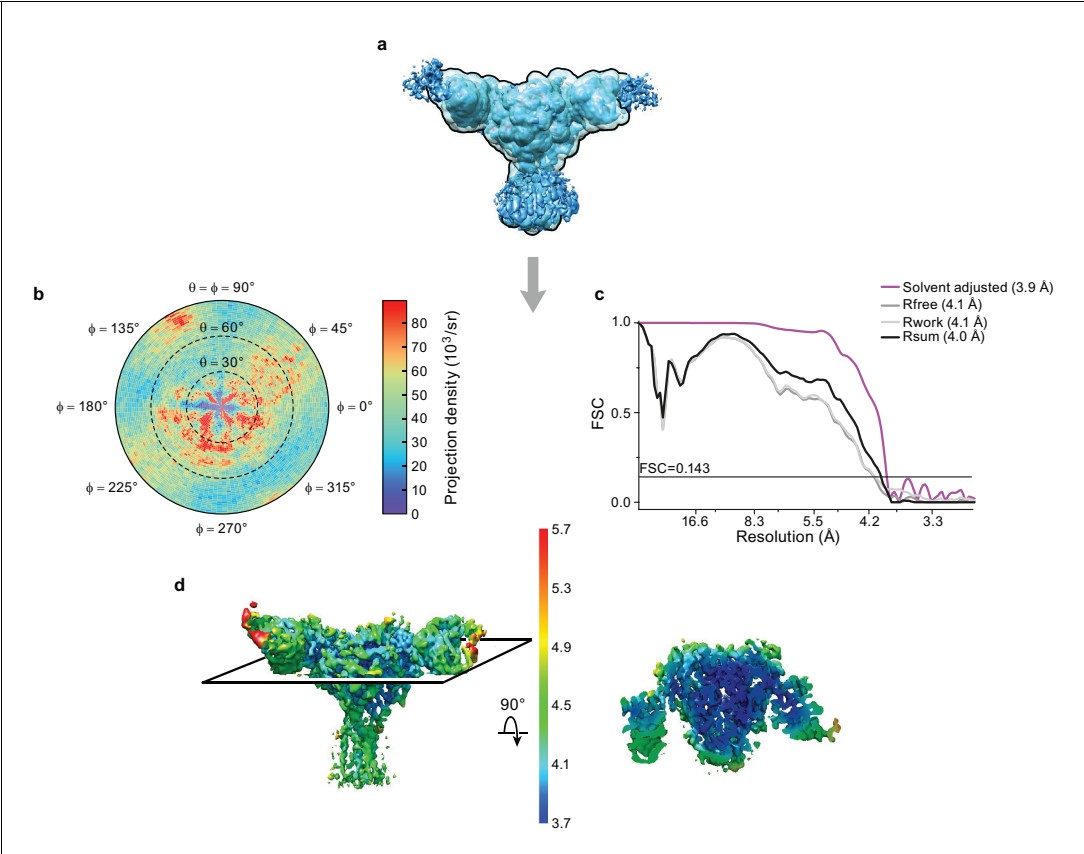

**Figure 3.** Cryo-EM analysis of final 3D reconstruction map. (**a**) Outline of mask used in the final 3D refinement of ΔENaC-7B1/10D4 complex. (**b**) Angular distribution of particle projections of the ΔENaC-7B1/10D4 complex. (**c**) Solvent adjusted FSC curve (purple) by cisTEM along with FSC curve between the atomic model of ΔENaC-7B1/10D4 complex and half map 1 ($R_{free}$ – dark grey), half map 2 ($R_{work}$ – light grey) and final reconstruction map ($R_{sum}$ – black). The solid line indicates FSC = 0.143. (**d**) 3D map colored according to local resolution estimation using Bsoft. Blue indicates regions where local resolution is estimated to be ~ 3.7 Å.

DOI: https://doi.org/10.7554/eLife.39340.016

The following source data and figure supplement are available for figure 3:

**Source data 1.** Statistics of data collection, three-dimensional reconstruction and model refinement

DOI: https://doi.org/10.7554/eLife.39340.018

**Figure supplement 1.** Cryo-EM data processing.

DOI: https://doi.org/10.7554/eLife.39340.017

ASIC, which was first illustrated in the crystal structure of chicken ASIC (cASIC) (*Jasti et al., 2007*) (chicken ASIC shares 18 – 20% sequence identity with human ENaC; *Figure 5*, *Figure 5—figure supplement 1*). Each subunit of ΔENaC harbors a cysteine-rich ECD resembling a hand with the palm, knuckle, finger, and thumb domains clenching a 'ball' of β strands. This compact organization is stabilized by eight disulfide bridges in the ECDs of α and γ and nine in β. Seven of the disulfide bonds are conserved throughout the ENaC/DEG family (*Figure 1—figure supplement 1*, *Figure 1—figure supplement 2*, *Figure 5a–c*). The eighth is unique to the three ENaC subunits. For the purpose of consistency in the following discussion, domain and secondary structure assignment in ENaC follow those of ASIC (*Figure 5d*).

At the center of the trimeric architecture of the ECD are β-sheets formed by β1, β3, β6, and β9-β12 that constitute the palm domain, which are divided into two sections, the upper and lower palm domains. The upper palm domain cradles the β-ball, which is composed of β2, β4, β5, β7, and β8 in all three subunits, contrary to previous findings which suggested that the α subunit lacked the β4 and β5 strands (*Stockand et al., 2008*). Completing the 'clench' around the β-ball are the α1 – 3 of the finger, α4 – 5 of the thumb, and α6 of the knuckle domains. The lower palm is directly linked to

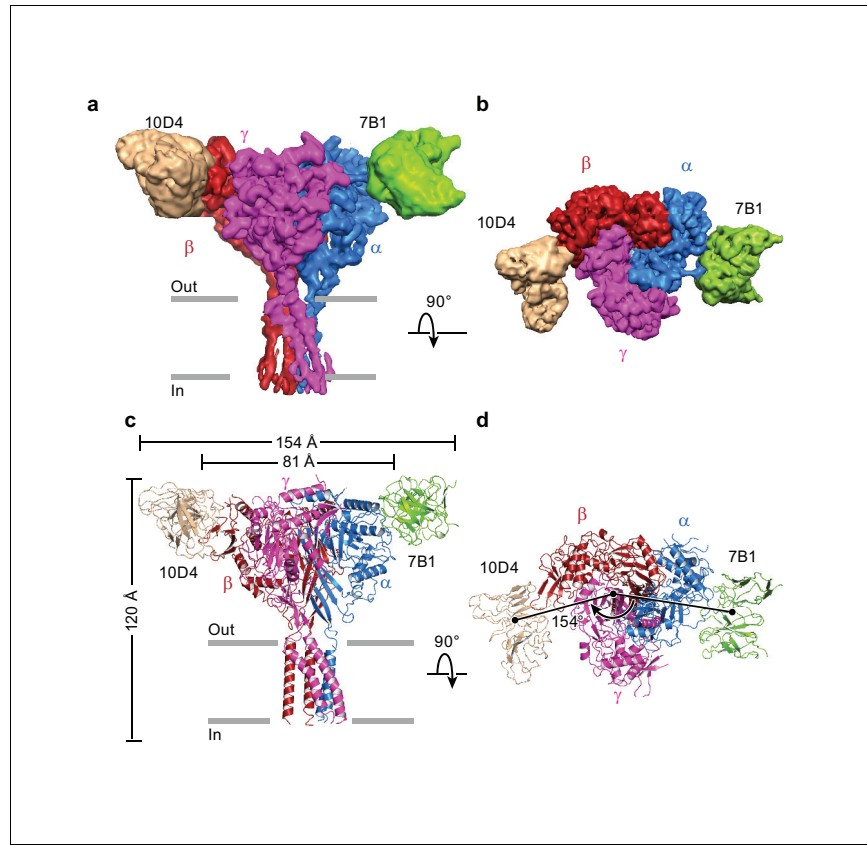

**Figure 4.** Architecture of the human epithelial sodium channel. (**a**) and (**b**), Cryo-EM map of the ΔENaC-7B1/10D4 complex viewed parallel (**a**) and perpendicular (**b**) to the membrane. The α, β, and γ subunits are colored blue, red, and magenta, respectively. The 7B1 and 10D4 Fv densities are colored green and wheat, respectively. (**c**) and (**d**), Cartoon representation of the ΔENaC-7B1/10D4 complex viewed and colored as in (**a**) and (**b**). The dimensions of the complex and ΔENaC alone are indicated. The centers of mass of the Fv's make 154° angle.
DOI: https://doi.org/10.7554/eLife.39340.019

The following video and figure supplements are available for figure 4:

**Figure supplement 1.** Unique map features facilitated subunit identification and model building.
DOI: https://doi.org/10.7554/eLife.39340.020

**Figure supplement 2.** Regions of the Δα ENaC subunit model shown in stick representation superimposed with the potential map in light grey mesh, contoured between 7.5 and 8.0 σ.
DOI: https://doi.org/10.7554/eLife.39340.021

**Figure supplement 3.** Regions of the Δβ ENaC subunit model shown in stick representation superimposed with the potential map in light grey mesh, contoured between 7.5 and 8.0 σ.
DOI: https://doi.org/10.7554/eLife.39340.022

**Figure supplement 4.** Regions of the Δγ ENaC subunit model shown in stick representation superimposed with the potential map in light grey mesh, contoured between 7.5 and 8.0 σ.
DOI: https://doi.org/10.7554/eLife.39340.023

**Figure supplement 5.** Select regions of the GRIP domain in all three subunits.
DOI: https://doi.org/10.7554/eLife.39340.024

**Figure 4—video 1.** Three dimensional architecture of the ENaC subunits.
DOI: https://doi.org/10.7554/eLife.39340.025

the transmembrane domain (TMD) via β1 and β12 and to the α4 and α5 of the thumb through β9 and β10. The thumb and the lower palm converge to forge interactions with the TMD at a juncture called the 'wrist' (**Figure 5—figure supplement 1**). Underscoring the importance of the wrist region and the critical roles that disulfide bridges play in maintaining the structural and functional integrity of ENaC, alterations of a conserved cysteine, α-Cys479 to an Arg, causes Liddle syndrome due to a missense mutation that not only eliminates a disulfide bridge located at the juncture of the thumb

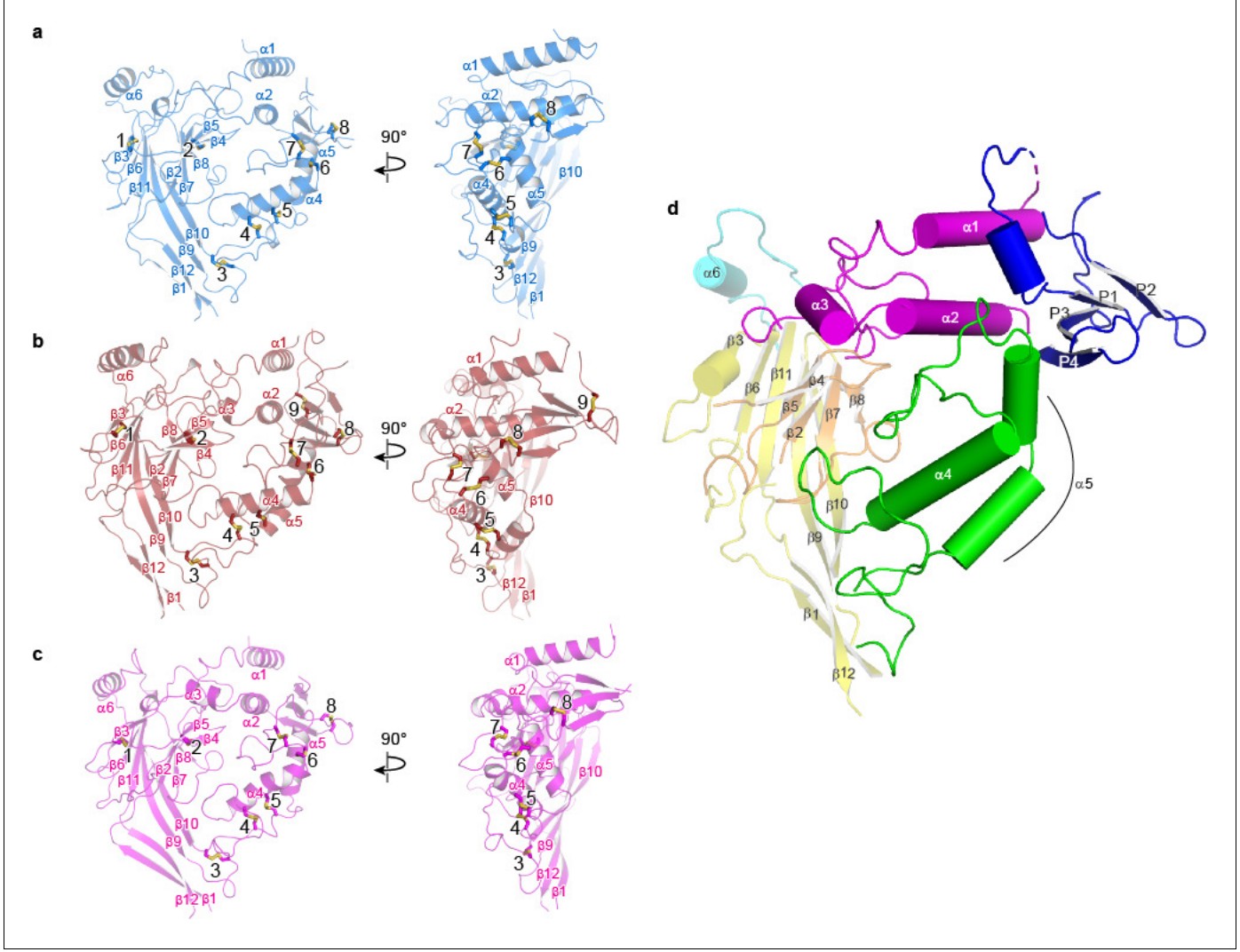

**Figure 5.** Domain organization in each subunit of ΔENaC resembles a hand clenching a ball. (a–c) Domain organization of each ENaC subunit and locations of disulfide bridges. Disulfide bridges 1–7 are conserved across ENaC/DEG family while the eighth disulfide bridge is shared by α (a), β (b), and γ (c) and located in the GRIP domain (P1 - P4). The β subunit contains a ninth disulfide bridge that is also located in the GRIP domain. All subunits are in cartoon representation and colored as in *Figure 4* and the disulfide bridges are in sticks representation. (d) Schematic diagram of secondary structure elements of ENaC subunits colored as follows: knuckle, cyan; palm, yellow; finger, purple; GRIP, blue; β-ball, orange; thumb, green.

DOI: https://doi.org/10.7554/eLife.39340.026

The following figure supplement is available for figure 5:

**Figure supplement 1.** Comparison between ENaC and ASIC subunit structure.
DOI: https://doi.org/10.7554/eLife.39340.027

and palm domains but also introduces a bulky, positively charged residue (*Salih et al., 2017*) (*Figure 5a*).

ENaC differs significantly from ASIC in both structure and primary sequence at the knuckle and finger domains (*Figure 1—figure supplement 1*, *Figure 1—figure supplement 2*, *Figure 5—figure supplement 1*). Each knuckle domain in ENaC makes extensive interactions with the α1 and α2 helices of the finger domain in the adjacent subunit (*Figure 6*). Together, the knuckle and finger domains of all three subunits form a 'collar' at the top of the ECD. Sequence alignment of the three subunits demonstrate divergence in amino acid sequence at the C-terminal end of both α1 and α6 in all three subunits, which results in distinct types of molecular interactions at each subunit interface that are, perhaps, associated with assembly and stability of the ENaC. The contact between the

finger and thumb domains is mediated by long antiparallel helices α1 and α2, which form a barrier between the thumb domain and the β6-β7 loop with α2 making the primary contacts with the thumb domain (*Figure 5—figure supplement 1*, *Figure 6*). The domain arrangement observed in the ΔENaC structure agrees with the functional study probing Na$^+$ binding sites in the α subunit of ENaC (*Kashlan et al., 2015*). The α2 helix makes an almost 90° turn towards the palm domain breaking the helix. This architecture marks another departure from ASIC, in which contacts between the finger and thumb domains are largely mediated by α1, α3, α5, and the α4-α5 loop (*Figure 5—figure supplement 1*).

The TMD is not well ordered, hampering our ability to model the entire TMD region and assign a functional state of the channel. Nevertheless, the EM map offers a glimpse of the positions of TM1 and TM2 on the extracellular side from each subunit (*Figure 6—figure supplement 1*). The overall configuration of the TMD shows that TM2 of all three subunits are positioned near the central axis, poised to mediate ion conduction in agreement with the crystal structures of ASIC and previous elegant functional studies probing ion selectivity and channel block (*Kellenberger and Schild, 2002*). Strikingly, the potential map for γ-TM1 on the extracellular side illustrates clear map for the main chain preceding the β1 strand validating a sequence disparity between the γ subunit and the other ENaC and ASIC subunits (*Figure 6—figure supplement 1*). The γ subunit lacks two residues preceding the palm domain (*Figure 6—figure supplement 1f*). Consequently, interactions within the wrist region in the γ subunit may differ from that of α and β subunits.

## The GRIP domain

Previous studies of ENaC have probed stretches of amino acids and their roles in ENaC function by perturbing known protease sites, observing changes in molecular weight, recording channel activity, and conducting cross-linking studies (*Bruns et al., 2007*; *Carattino et al., 2006*). The structure of ΔENaC indicates that these stretches of 20 – 40 amino acids are pieces of larger domains located in the periphery of the ECD near subunit interfaces (*Figure 7*). These stretches of amino acids, located between α1 and α2 are unique to ENaC and are responsible for channel G̲ating R̲elief of I̲nhibition by P̲roteolysis and will hereafter be referred to as the GRIP domain. Each GRIP domain is composed of a core of β strands that forge interactions with the finger and thumb domains forming a β-sheet 'blanket' that conceals the α2 helix of the finger (*Figure 7—figure supplement 1*). Surprisingly, although the β subunit is not known to gate the channel via proteolysis, it also possesses a GRIP domain with similar organization to those of the α and γ subunits.

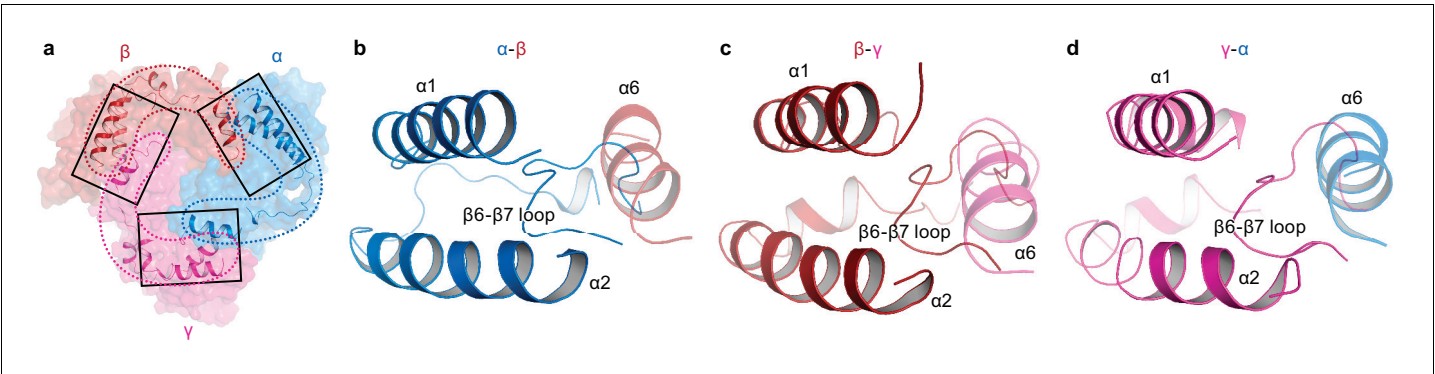

**Figure 6.** Intersubunit interactions in ΔENaC in the finger and knuckle domains. (a) The finger and knuckle domains forge intersubunit interactions forming a 'collar' at the top of the ECD. Surface representation of ΔENaC viewed perpendicular to the membrane. Subunits are colored as in *Figure 4*. The finger (α1 – 3) and knuckle domains (α6) are shown in cartoon representation. (b–d), Detailed view of the interfaces boxed in (a). The views are parallel to the membrane and show how the helices from the finger and knuckle domains constitute an enclosure around the β6-β7 loop.
DOI: https://doi.org/10.7554/eLife.39340.028
The following figure supplement is available for figure 6:

**Figure supplement 1.** The ΔENaC structure demonstrates asymmetric interactions at the wrist region.
DOI: https://doi.org/10.7554/eLife.39340.029

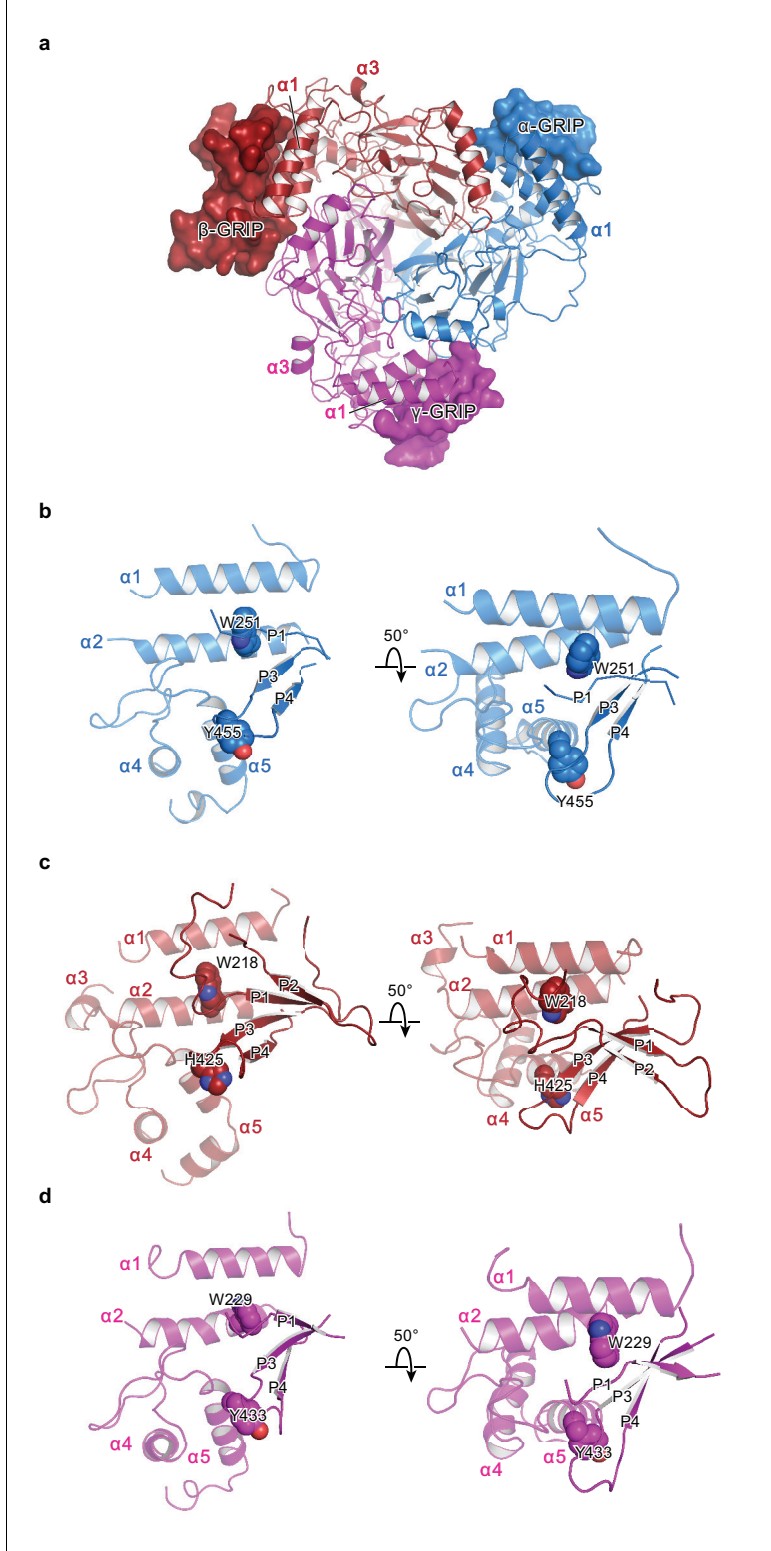

**Figure 7.** The protease-sensitive domain in ENaC is part of the GRIP domain. (**a**) ΔENaC is shown in cartoon representation and colored as in *Figure 4*. The GRIP domain is shown in surface representation. Close-up view of the cleft formed by the finger and thumb domains and the P3-P4 segments in α (**b**), β (**c**), and γ (**d**). The cleft is occupied by the P1 segment of the GRIP domain. All three subunits contain conserved tryptophans in α2 (αTrp251, βTrp218, and γTrp229), which interacts with the P1 segment. The P3 and P4 strands are stabilized by the α5 of the thumb domain by docking on top of aromatic residues (αTyr455, βHis425, and γTyr433).

*Figure 7 continued on next page*

*Figure 7 continued*

DOI: https://doi.org/10.7554/eLife.39340.030

The following figure supplement is available for figure 7:

**Figure supplement 1.** Cryo-EM map of the P1 segments in α, β, and γ demonstrates critical interactions with the finger and thumb domains.

DOI: https://doi.org/10.7554/eLife.39340.031

In all three subunits, the GRIP domains comprise two antiparallel β strands stapled together by a disulfide bond located in the loop that rests against the thumb domain (*Figure 7b–d*, *Figure 7—figure supplement 1b–d*). Furthermore, an additional disulfide bond in the loop near the β-γ interface stabilizes the GRIP domain of the β subunit. We suspect that this additional disulfide bond contributes to the well-ordered behavior of the β GRIP domain, allowing the resolution of nearly the whole segment between α1 and α2 in the β subunit. Moreover, the 10D4 Fab binds the β GRIP domain, allowing us to resolve two additional antiparallel β strands. In the α and γ subunits, we can only identify one stretch of residues that adopt an extended conformation. Based on the shared features observed in all three subunits, it is plausible that the α and γ subunits also contain a fourth β strand. With four possible β strands in the GRIP domains, each strand or stretch of peptides is referred to here as P1-4 (*Figure 7*).

Structural insight gleaned from the β GRIP domain reveals the possible positions of the functionally well-characterized but structurally elusive inhibitory tracts and furin and other protease sites in the α and γ GRIP domains. Studies by the Kleyman group have identified 8- and 11-mer peptide tracts within the α and γ GRIP domains, respectively, which are implicated in channel gating (*Passero et al., 2010*; *Carattino et al., 2008b*; *Kashlan et al., 2010*). Sequence comparison between the three subunits suggests that the inhibitory tracts contain the P1 strand (*Figure 7b – d*, *Figure 7—figure supplement 1b–d*, *Figure 1—figure supplement 1*). Based on this configuration, the first furin site lies at the N-terminal side of P1 whereas the second protease site (furin for α and other protease sites for γ) is likely located at the C-terminal side of P2. The anti-parallel organization of P1 and P2 strands places the two protease sites in close proximity to each other. We speculate that this arrangement allows for efficient proteolysis, especially for the cleavage of the α subunit by furin.

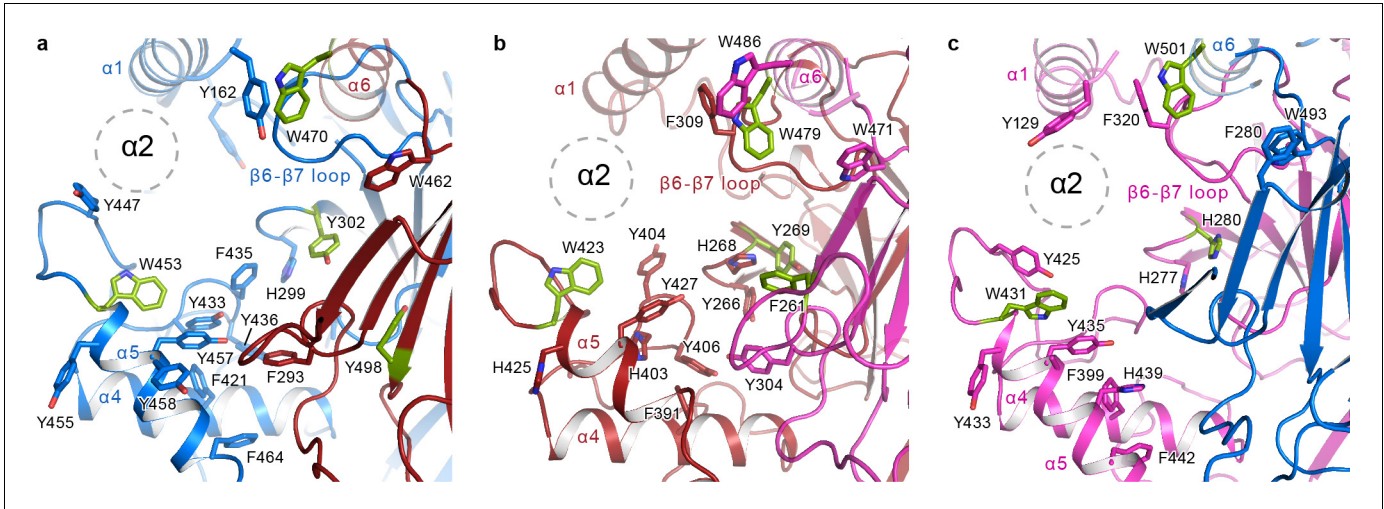

**Figure 8.** The α2 helix is buried in the aromatic pocket formed by key gating domains in ENaC. The aromatic pockets at the α-β (a), β-γ (b), and γ-α (c) interfaces are shown in cartoon representation. The aromatic residues are shown in sticks representation. Residues conserved in ASIC are colored green. The α2 helices and the GRIP domains are omitted for clarity. The position of the α2 is shown as dotted circles.

DOI: https://doi.org/10.7554/eLife.39340.032

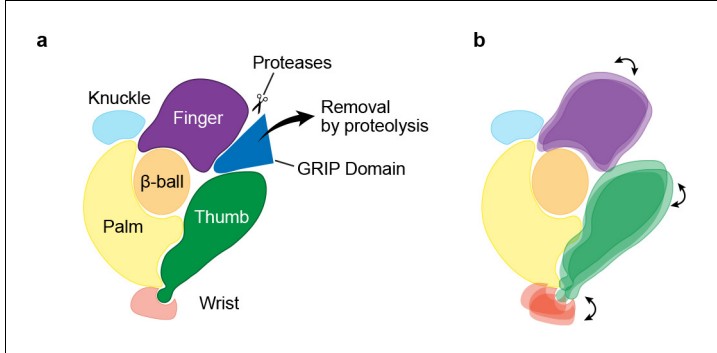

**Figure 9.** Mechanism of protease-dependent gating in a single ENaC subunit. Removal of the protease sensitive segments of the GRIP domain (**a**) induces conformational changes in the finger and thumb domains (**b**), which is perhaps coupled to ion channel gating through the wrist.
DOI: https://doi.org/10.7554/eLife.39340.033

## Aromatic pocket

The first crystal structure of ASIC identified the finger and the thumb domains as major players in ion channel gating. Rearrangements of these domains are coupled to the TMD via the wrist. Additionally, the crystal structure provided insight into the domain essential for fine-tuning ASIC pH-response, deemed the acidic pocket, formed by the β-ball, finger, and thumb domains of one subunit, and the palm domain of the adjacent subunit (*Vullo et al., 2017*). While the acidic pockets in ASIC are lined with negatively charged residues, the equivalent crevices in ENaC are replete with aromatic residues. In fact, aside from Ser428 of Δβ (Asp346 in ASIC), the equivalent sites in the thumb domain that are acidic in ASIC are occupied by tyrosines in all three subunits (*Figure 8*, *Figure 1—figure supplement 1*, *Figure 1—figure supplement 2*) (*Jasti et al., 2007*). Accordingly, the pocket that is largely occupied by α2 in ΔENaC is referred to here as the aromatic pocket.

Tucked in the aromatic pocket, α2 makes contacts with all critical elements of the gating machinery in ENaC. This observation is consistent with studies finding that site-directed mutagenesis perturbing residues in the α2 results in changes in $Na^+$ self-inhibition and binding of the P1 segment of the GRIP domain (*Kashlan et al., 2010*). In all three ENaC subunits, the α2 forms contacts with the thumb, α1 helix, the β6-β7 loop and the GRIP domain, and with the knuckle and the upper palm domains in the adjacent subunit (*Figure 8*). Moreover, studies using synthesized 8-mer (LPHPLQRL) and 11-mer peptides (RFLNLIPLLVF) (*Passero et al., 2010*), the inhibitory peptides of α and γ subunits, respectively, have identified residues in α2 to be critical to the binding of the inhibitory peptides. These peptides pack against α2 and form a wedge between the thumb domain and α1 in the ΔENaC structure (*Figure 7b–d* and *Figure 7—figure supplement 1b–d*). These inhibitory peptides contain prolines that introduce a kink within the tract that may serve as a point that divides P1 into two segments: the N-terminal side, which interacts with the finger and thumb; and the C-terminal side, which interacts primarily with α2 and P3. The observed orientation of the P1 segment is consistent with the cross-linking experiments by Kashlan et al., which provided two major findings: (1) the inhibitory tracts adopt an extended conformation and  (2) the N-terminal side of the peptide binds near the thumb/finger interface (*Kashlan et al., 2010*; *Kashlan et al., 2012*).

The potential map for α-P1 suggests that the N-terminal side mirrors that of β-P1 forging contacts with the α1 helix (*Figure 7—figure supplement 1b,c*). In contrast, the potential map for the γ-P1 suggests that the peptide interacts with the thumb and α1/α2 more extensively and extends toward α3 (*Figure 7—figure supplement 1d*). These distinct points of contact with the finger and thumb domains between the α- and γ-P1 segments may influence the extent to which the subunits influence channel $P_o$. While removal of the inhibitory tract in α transitions the channel to an intermediate $P_o$ state, excision of the γ-P1 segment places the channel in the high $P_o$ state; this high $P_o$ state can be accomplished without the removal of the α-P1 (*Carattino et al., 2008a*). The visual evidence of direct interactions between P1 and the finger and thumb domains demonstrated in the ΔENaC structure sheds light into how these inhibitory tracts can modulate channel function in ENaC.

## Mechanism

ΔENaC follows a common organization that was first observed in ASICs: a scaffolding structure in the upper palm, the flexible lower palm which is tethered to the TM and thumb, and the β-ball (*Figure 5—figure supplement 1*) (*Jasti et al., 2007*). However, the specialized finger domains deviate from what is observed in ASIC and such deviations accommodate the distinct functions between the proton sensors of ASIC and the protease-sensitive regulators of ENaC.

Key gating structures are preserved, albeit with specific structural configurations in both ASIC and ENaC supporting the idea that the superfamily of ENaC/DEG channels conform to a gating scheme that involves conformational changes of the finger and thumb domains, rearrangements that are propagated to the ion channel via the wrist (*Jasti et al., 2007*). In the case for ENaC, a speculative model for gating involves proteolysis and the subsequent removal of the P1 segment, which serves as a wedge, inducing rearrangements of the finger and thumb domains (*Figure 9*).

The structural work presented here provides new insight into ENaC assembly and gating. The structure unveils the positions of the GRIP domains, specifically the key peptidyl tracts that inhibit ENaC activity, and the distinct interactions that they mediate with the finger and thumb domains. Furthermore, it reveals that there are different interactions between the finger and knuckle domains at each subunit interface, and between the base of the thumb and the TMD in the wrist region suggesting that each subunit differentially contributes toward gating the channel, supporting electrophysiological findings. Importantly, the structure provides the first molecular model for protease-dependent regulation of ENaC opening and $Na^+$ and water homeostasis.

# Materials and methods

### Key resources table

| Reagent type (species) or resource | Designation | Source or reference | Identifiers | Additional information |
|---|---|---|---|---|
| Gene (*Homo sapiens*) | amiloride-sensitive sodium channel subunit alpha isoform 1 | Synthetic | NCBI Reference Sequence: NP_001029.1 | Gene sysnthesized by BioBasic |
| Gene (*Homo sapiens*) | amiloride-sensitive sodium channel subunit beta | Synthetic | NCBI Reference Sequence: NP_000327.2 | Gene sysnthesized by BioBasic |
| Gene (*Homo sapiens*) | amiloride-sensitive sodium channel subunit gamma | Synthetic | NCBI Reference Sequence: NP_001030.2 | Gene sysnthesized by BioBasic |
| Cell line (*Homo sapiens*) | HEK293S GnTI- | ATCC | Cat # ATCC CRL-3022 | |
| Cell line (*Homo sapiens*) | HEK293T/17 | ATCC | Cat # ATCC CRL-11268 | |
| Antibody | 7B1 | OHSU VGTI, Monoclonal Antibody Core | AB_2744525 | Isotype IgG2a, kappa |
| Antibody | 10D4 | OHSU VGTI, Monoclonal Antibody Core | AB_2744526 | Isotype IgG1, kappa |
| Recombinant DNA reagent | pEG BacMam | Gift from Eric Gouaux | doi: 10.1038/ nprot.2014.173 | |
| Commercial assay or kit | mMessage mMachine T7 Ultra Transcription | Ambion | AM1345 | |
| Chemical compound, drug | Amiloride hydroschloride hydrate | Sigma | A7410 | |
| Chemical compound, drug | Phenamil mesylate | Tocris | Cat # 3379 | |
| Software, algorithm | Relion-2.0 | doi: 10.1016/ j.jsb.2012.09.006 | RRID:SCR_016274 | https://www2.mrc-lmb.cam.ac.uk/relion/index.php?title=Main_Page |

*Continued on next page*

*Continued*

| Reagent type (species) or resource | Designation | Source or reference | Identifiers | Additional information |
|---|---|---|---|---|
| Software, algorithm | DoG picker | doi: 10.1016/j.jsb.2009.01.004 | OMICS_27772 | https://omictools.com/dog-picker-tool |
| Software, algorithm | MotionCor2 | doi:10.1038/nmeth.4193 | SCR_016499 | http://msg.ucsf.edu/em/software/motioncor2.html |
| Software, algorithm | Gctf | doi:10.1016/j.jsb.2015.11.003 | SCR_016500 | https://www.mrc-lmb.cam.ac.uk/kzhang/Gctf/ |
| Software, algorithm | cryoSparc | doi:10.1038/nmeth.4169 | SCR_016501 | https://cryosparc.com/ |
| Software, algorithm | cisTEM | https://doi.org/10.1038/nmeth.4672 | SCR_016502 | https://cistem.org/ |
| Software, algorithm | Bsoft | doi:10.1006/jsbi.2001.4339 | SCR_016503 | https://lsbr.niams.nih.gov/bsoft/ |
| Software, algorithm | Pymol | PyMOL Molecular Graphics System, Schrŝdinger, LLC | RRID:SCR_000305 | http://www.pymol.org/ |
| Software, algorithm | UCSF Chimera | doi:10.1002/jcc.20084 | RRID:SCR_004097 | http://plato.cgl.ucsf.edu/chimera/ |
| Software, algorithm | Coot | https://doi.org/10.1107/S0907444904019158 | RRID:SCR_014222 | https://www2.mrc-lmb.cam.ac.uk/personal/pemsley/coot/ |
| Software, algorithm | Rosetta | https://doi.org/10.1371/journal.pone.0020450 | RRID:SCR_015701 | https://www.rosettacommons.org/ |
| Software, algorithm | Phenix | doi:10.1107/S2059798318006551 | RRID:SCR_014224 | https://www.phenix-online.org/ |
| Software, algorithm | Jpred4 | https://doi.org/10.1093/nar/gkn238 | SCR_016504 | www.compbio.dundee.ac.uk/jpred/ |
| Software, algorithm | Psipred v3.3 | https://doi.org/10.1093/bioinformatics/16.4.404 | RRID:SCR_010246 | www.bioinf.cs.ucl.ac.uk/psipred |
| Software, algorithm | QUARK | doi:10.1002/prot.24065 (2012) | OMICS_10835 | https://omictools.com/quark-tool |
| Software, algorithm | Molprobity | doi:10.1107/S0907444909042073 | RRID:SCR_014226 | http://molprobity.biochem.duke.edu |

## Construct design

The cDNA encoding the full length α, β and γ subunits of human ENaC were cloned into pEG Bac-Mam expression vector harboring an N-terminal eGFP (*Goehring et al., 2014*). The Δα was generated by removing both N- and C-terminal segments and modifying the furin sites obtaining a mutant variant of the α subunit lacking 42 and 89 residues at N- and C- termini, respectively. While the Δβ was designed to possess truncations of 30 and 79 residues at the N- and C-terminal regions, $Δβ_{ASIC}$ has the same truncated regions as Δβ but contains an additional N-terminal 2 – 22 residues of cASIC upstream of Δβ. Lastly, Δγ lacks 20 and 79 residues at the N- and C-terminal domains, has modified furin and prostasin sites, and includes a Strep-tag II, an octa-histidine tag, eGFP, and Thrombin cleavage site at the N-terminus.

## Generation and isolation of Fabs

Mouse monoclonal antibodies 7B1 and 10D4 were generated using standard procedure by Dan Cawley at the Vaccine and Gene Therapy Institute (OHSU). Liposomes containing asolectin:cholesterol:lipidA:brain polar lipid extract (BPLE) (16:4.6:1:5.3) were prepared in 20 mM Tris, 150 mM NaCl at pH 8.0 at a concentration of 40 mg/ml. The mixture was subjected to repeated freeze-thaw cycles followed by extrusion through a 200-nm filter. Purified $ΔENaC_{ASIC}$ (Δα, $Δβ_{ASIC}$, Δγ) protein was added to the liposome mixture in the presence of 400 mM NaCl and 0.8% Na-cholate and passed through a PD-10 desalting column to remove excess salt and detergent. Mice were immunized with approximately 30 μg of the reconstituted $ΔENaC_{ASIC}$ for generation of hybdridoma cell lines

(*Figure 1—figure supplement 3*). Monoclonal antibodies were screened by FSEC and BioDot blot to identify clones that recognize tertiary or primary epitopes. The 7B1 and 10D4 mAbs were selected because they recognize tertiary epitopes of ENaC. The mAbs were purified, and their Fabs were generated by papain cleavage. Fab 7B1 was isolated by anion exchange using HiTrap Q HP column while Fab 10D4 was eluted using Protein A column to remove Fc. After isolation, both Fabs were dialyzed in 200 mM NaCl and 20 mM Tris at pH 8.0.

## Expression and purification of ΔENaC-Fab complexes

Human embryonic kidney cells lacking N-acetylglucosaminyltransferase I (HEK293S GnTI⁻ cells) were grown in suspension at a density of $2-4 \times 10^6$ cells / ml in Freestyle medium with 2% FBS and trans-duced with the virus (Δα, Δβ and Δγ) at a multiplicity of infection (MOI) of 1 and incubated at 37°C. Eight hours post-transduction, sodium butyrate and phenamil mesylate were added to 10 mM and 500 nM, respectively, and cells were incubated at 30°C. After 36 hr, the cells were collected by cen-trifugation at 4790 xg for 15 min. The pellet was washed with 20 mM Tris, 200 mM NaCl and fol-lowed by a second round of centrifugation at 4790 xg for 15 min. Cells were homogenized with a dounce homogenizer and sonicated in 20 mM Tris, 200 mM NaCl, 5 mM $MgCl_2$, 25 µg/ml DNase I and protease inhibitors. Lysed cells were centrifuged at 9715 xg for 20 min; the resulting superna-tant containing the membrane fraction was further centrifuged at 100,000 xg for 1 hr. Membrane pellets were resuspended and solubilized in 20 mM TRIS pH 8, 200 mM NaCl, 20 mM n-dodecyl-β-D-maltopyranoside (DDM, Anatrace), 3 mM cholesteryl hemisuccinate (CHS), 2 mM ATP, 2 mM $MgSO_4$, protease inhibitor and 25 U/mL nuclease for 1 hr at 4°C. The solubilized fraction was iso-lated by ultracentrifugation 100,000 xg for 1 hr, and ΔENaC was bound to streptactin resin packed into an XK-16 column. The column was washed with 20 mM TRIS, 200 mM NaCl, 0.5 mM DDM, 75 µM CHS and 25 U/mL nuclease, followed by an additional wash of the same buffer containing 2 mM ATP, and eluted with 2.5 mM desthiobiotin. The eluted fractions were concentrated and then incu-bated with either one Fab 10D4 (monoFab complex) or two Fabs 7B1 and 10D4 (diFab complex) in a 1:3 molar ratio of ENaC:Fab for 10 min, and clarified by ultracentrifugation 100,000 xg for 1 hr. The supernatant was injected onto a Superose 6 Increase 10/300 GL column equilibrated in 20 mM TRIS pH 8.0, 200 mM NaCl, 0.5 mM DDM, 75 µM CHS and 1 mM TCEP to isolate the protein com-plex by size-exclusion chromatography. Monodispersed fractions were pooled and concentrated to 2.2 mg/mL.

For FSEC experiments analyzing peak shifts of the ΔENaC and FL-ENaC with 7B1 and 10D4, ΔENaC was expressed in HEK 293S GnTI⁻, as described above, while FL-ENaC was expressed in HEK 293T/17. The HEK 293T/17 cells were grown in suspension at a density of $2-4 \times 10^6$ cells / ml in Freestyle medium with 2% FBS and transduced with the virus (FL-α, FL-β and FL-γ) at a multiplicity of infection (MOI) of 1 and incubated at 37°C. Eight hours post-transduction, 500 nM phenamil mesy-late was added, and cells were incubated at 30°C. After 36 hr post-transduction, the cells were col-lected by centrifugation at 4790 xg for 15 min. The pellet was washed with 20 mM Tris, 200 mM NaCl and followed by a second round of centrifugation at 4790 xg for 15 min. Cell pellets were resuspended and solubilized in 20 mM TRIS pH 8, 200 mM NaCl, 20 mM n-dodecyl-β-D-maltopyra-noside (DDM, Anatrace), 3 mM cholesteryl hemisuccinate (CHS), protease inhibitor and 25 U/mL nuclease for 1 hr at 4°C. The solubilized fraction was isolated by ultracentrifugation 100,000 xg for 1 hr, then incubated with either one Fab (7B1 or 10D4) or two Fabs (7B1/10D4) in a 1:3 molar ratio of ENaC:Fab for 10 min, and clarified by ultracentrifugation 100,000 xg for 1 hr. The supernatant was injected onto a Superose 6 Increase 10/300 GL column for FSEC analysis.

## Immunoblotting

Aliquots of 7 µg purified ENaC were incubated with 2.5 µg/mL trypsin for 10 min at room tempera-ture. These samples were then run through 4 – 20% Criterion SDS-PAGE gels and blotted onto nitro-cellulose membranes according to manufacturer's instructions (Bio-Rad). After blocking overnight in 5% non-fat milk in TBS, membranes were incubated in primary antibody (ENaC α subunit, 6 µg/blot SC-21012; ENaC β subunit, 6 µg/blot SC-21013; ENaC γ subunit, 10 µg/blot abcam ab133430) for 2 hr. The membranes were then incubated in 1 µg/blot IRDye 800CW goat anti-rabbit IgG (Licor) for 1 hr.

## Image acquisition and processing

Purified ΔENaC-Fab complexes were applied to glow-discharged Quantifoil holey carbon grids (Au 1.2 μm/1.3 μm hole space/hole separation, 300mesh), blotted using a Vitrobot Mark III (FEI) with the following conditions, 7 s wait time, and 5 s blot time at 100% humidity, and then plunge-frozen in liquid ethane cooled by liquid nitrogen. All images were collected on a Titan Krios electron microscope operating at 300 kV at the Multiscale Microscopy Core (OHSU). Images were recorded by a Gatan K2 Summit direct electron detector operating in super-resolution mode, and the images were collected using the automated acquisition program SerialEM (*Mastronade, 2003*). Magnification of the recorded images corresponded to a pixel size of 1.33 Å in counting mode (0.665 Å in super-resolution mode). For the ΔENaC-10D4 complex, two data sets were acquired and were initially processed separately, and subsequently combined for 3D reconstruction. Each image in the first dataset was dose-fractionated to 30 frames with 0.5 s per frame and a total exposure time and dose of 15 s and 54 e$^-$/Å$^2$, respectively. The second dataset was collected in counting mode, and was therefore not binned when combined with the first where each image was dose-fractionated to 60 frames with 0.25 s per frame and a total exposure time and dose of 15 s and 50 e$^-$/Å$^2$, respectively. Similarly, two separate datasets were obtained for ΔENaC-7B1/10D4 complex in super-resolution mode. Like in the monoFab complex, each data set was processed separately and later combined for further analysis and 3D reconstruction. The images of the first dataset of the diFab complex were dose-fractionated to 40 frames with 0.25 s per frame and a total exposure time and dose of 10 s and 62 e$^-$/Å$^2$, respectively, while the images of the second dataset were dose-fractionated to 48 frames with 0.25 s per frame and a total exposure time and dose of 12 s and 71 e$^-$/Å$^2$, respectively.

The ΔENaC-10D4 data set collected in super-resolution mode was binned 2 × 2 while the ΔENaC-10D4 data set collected in counting mode was left unbinned. Both data sets were motion corrected using MotionCor2 (*Zheng et al., 2017*), and automated particle selection was performed using DoGPicker (*Voss et al., 2009*). Defocus values for individual particles were estimated using Gctf (*Zhang, 2016*), and particles belonging to low-abundance classes were removed via 2D classification and 3D classification in RELION (*Scheres, 2012*). The final set of particles was further analyzed in cryoSPARC and refined to a nominal resolution of 5.4 Å (*Punjani et al., 2017*).

For the ΔENaC-7B1/10D4 data sets, super-resolution counting images were 2 × 2 binned, and motion corrected using MotionCor2. Manual and automated particle selections were performed where DoGPicker was utilized for the latter resulting in a total of 667,984 particles. Defocus values for individual particles were estimated using Gctf, and particles of low-abundance classes via 2D classification in RELION were removed. For 3D classification in RELION, a reference model of a low-resolution map of ENaC-7B1/10D4 obtained from a data set (14.4 Å) was low-pass filtered to 50 Å, and particles were classified into two classes where the major class contained 385,997 particles. Duplicates (as a result of RELION2.0 re-centering particles after 2D classification) and particles close to micrograph edges were removed, resulting in 329,180 particles that were subjected to *ab initio* 3D classification in cryoSPARC (*Punjani et al., 2017*), and 3D classification and refinement in cisTEM (*Grant et al., 2018*). Particles belonging to the low abundance class in cryoSPARC and cisTEM were discarded yielding 244,223 and 290,007 particles, respectively. Using default settings in cryoSPARC, particles with class probability of > 0.9 were used for refinement; thus, final reconstruction and refinement used 244,223 particles. For cisTEM, initial 3D classification and refinement was done using a refinement threshold of 8 Å and applying a mask during the last few iterations that excluded the constant domain (Fc) of the Fabs. During this process, we noticed that extraneous features, such as the micelle, were having a strong influence on alignment and classification, so the cisTEM particles were then re-processed using a mask that excluded both the micelle and the Fc of the Fabs, and aligned with a 5.4 Å limit. This dataset consisted of 302,263 particles and improved the resolution, as determined by the FSC = 0.143 criterion (~3.9 Å). More importantly, the electrostatic potential map was notably improved in the regions of interest. The resolutions reported in *Figure 3—source data 1* are based on the FSC = 0.143 criterion (gold-standard in the case of RELION and cryoSPARC). Final resolution reported in *Figure 3—source data 1* are solvent adjusted FSC = 0.143 criterion. No symmetry was applied during data processing.

## Model building

Homology models of the human α, β, and γ subunits were generated with the crystal structure of cASIC (*Jasti et al., 2007*) (PDB code: 2QTS, chain A) as a template using SWISS-MODEL server and homology models for the Fabs were also generated by SWISS-MODEL (*Arnold et al., 2006*). All models were docked into the EM potential in UCSF Chimera then rigid-body fitted into the EM potential using Coot (*Pettersen et al., 2004*; *Emsley and Cowtan, 2004*). We incorporated models generated from Rosetta (*DiMaio et al., 2011*) into manual fitting and adjustments during model-building in Coot to build the palm, knuckle, TM, thumb, β-ball, and finger domains. To build the GRIP domains, we integrated analysis from Jpred4, PSIPRED v3.3, and QUARK online *ab initio* protein structure prediction to support our analysis of the cryo-EM map (*Buchan et al., 2013*; *Xu and Zhang, 2012*; *Drozdetskiy et al., 2015*). The final model was subjected to refinement using the module phenix.real_space_refine in PHENIX (*Adams et al., 2011*).

The cryo-EM map in all three subunits preceding the N-terminal side of the α2 helices was unambiguous and showed features consisting of two β-strands connected by a loop, the P3 and P4 segments of the GRIP domains (*Figure 4—figure supplement 5*). Secondary structure prediction analysis by online servers Jpred4, PSIPRED v3.3, and QUARK supported this observation (*Buchan et al., 2013*; *Xu and Zhang, 2012*; *Drozdetskiy et al., 2015*). We found that the potential map in the β subunit had the best-defined feature demonstrating four β strands (*Figure 4—figure supplement 5*). Based on the cryo-EM map, the P1 segments in α and γ adopt β strand-like conformations, like in the β subunit, which is also supported by the secondary structure prediction servers (*Figure 4*, *Figure 4—figure supplement 5*). The regions between P1 and P3 in the α and γ subunits, however, are disordered in the cryo-EM map. We built stretches of residues into the P1 potential maps in the α- and γ-GRIP domains using sequence alignment with the β-GRIP domain and cryo-EM potential map features as guides.

For validation, FSC curves were calculated between the final model and the EM map as well as the two half maps generated by cisTEM. We implemented MolProbity to analyze geometries of the atomic model (*Chen et al., 2010*). All figures of map and atomic model were prepared using UCSF Chimera and Pymol (*Pettersen et al., 2004*).

## Two-electrode voltage clamp electrophysiology

All constructs used for two-electrode voltage clamp electrophysiology (TEVC) experiments were cloned into pGEM vector, linearized and transcribed to mRNA using mMESSAGE mMACHINE T7 Ultra Kit (Ambion) procedure. *Xenopus laevis* oocytes purchased from Ecocyte were injected with a volume of 50 nL containing either 0.5 – 1.0 ng of each FL-ENaC subunit mRNA or 5 ng of each ΔENaC and Δ*ENaC subunit mRNA. For experiments containing combinations of FL-ENaC and ΔENaC, 5 ng of each subunit mRNA was injected. Oocytes were incubated at 16°C for 12 – 48 hr in the presence of 100 μM amiloride and 250 μg/mL amikacin. The recordings were performed using two different ionic solutions with or without 100 μM amiloride (110 mM KCl and 110 mM NaCl) where all buffers additionally contained 1.8 mM $CaCl_2$ and 10 mM HEPES (pH 7.4). Macroscopic ENaC currents are defined as the difference between inwards currents obtained in the absence and in the presence of 100 μM amiloride. To test full activation of ΔENaC constructs, 2.5 μg/mL Trypsin was perfused for 5 min in the presence of 100 μM amiloride. Amiloride-sensitive currents were recorded prior to Trypsin treatment as well as after in order to determine the increase in current amplitude. All recording experiments were carried out at a holding potential of −60 mV and repeated independently at least three times.

## Whole-cell patch clamp electrophysiology

HEK293S GnTI⁻ cells were grown in suspension at a density of $2-4 \times 10^6$/ml in Freestyle medium with 2% FBS and transduced with the virus (Δα, Δβ, and Δγ; or FL-α, FL-β, and FL-γ) at a multiplicity of infection (MOI) of 1 and incubated in the presence of 500 nM phenamil mesylate at 30°C for 12 hr. Five hours before recording, cells were transferred to wells containing glass coverslips at a density of $0.3 – 0.5 \times 10^6$ cells/ml and in Dulbecco's Modified Eagle Medium supplemented with 5% FBS and 500 nM phenamil mesylate. Whole-cell recordings were carried out 17 – 24 hr after transduction. Pipettes were pulled and polished to 2 – 2.5 MΩ resistance and filled with internal solution containing (in mM): 150 KCl, 2 $MgCl_2$, 5 EGTA and 10 HEPES (pH 7.35). External solution contained

(in mM): 150 NaCl, 2 MgCl$_2$, 2 CaCl$_2$, 10 HEPES (pH 7.4), and 0.1 amiloride. Test external solution did not contain 0.1 mM amiloride. As in TEVC experiments, macroscopic ENaC currents are defined as the difference between inwards currents obtained in the absence and in the presence of 100 μM amiloride. Holding potential was at −60 mV.

### Confocal microscopy

Six mg of mAb 10D4 was dialyzed into 0.2 M carbonate-bicarbonate (Na$_2$CO$_3$/NaHCO$_3$) solution buffered at pH 9.0. The dialyzed mAb was concentrated to 6 mg/mL. Tetramethylrhodamine (TRITC, ThermoFisher 46112) was dissolved in DMSO at a final concentration of 1 mg/mL. To the 10D4 solution, 35 μg of TRITC was slowly added and mixed thoroughly. The 10D4-TRITC mix was incubated at room temperature in the dark for 2 hr followed by gel filtration to remove excess TRITC. The carbonate-bicarbonate buffer was exchanged with Tris-buffered saline buffer (200 mM NaCl, 20 mM TRIS, pH 8.0) using PD-10 desalting column. The dye:protein molar ratio of the final TRITC-labeled mAb 10D4 in TBS buffer was approximately 2.8.

HEK293S GNTI$^-$ cells were resuspended from DMEM into 2 mL HBSS media, stained with 10 μg (5 μg/μL stock) of WGA Alexa Fluor 647 conjugate (ThermoFisher W32466) and 170 μg (4.9 μg/μL stock) of 10D4-TRITC and subsequently incubated at 37°C for 10 min. The cells were then washed with PBS two times before resuspended in 1 mL HBSS. Live cell imaging was performed on a Yokogawa CSU-W1 spinning disk confocal microscope using a 60 × 1.4 Plan Apo VC objective. Images were acquired at a pixel size of 0.108 μm for three different wavelengths, starting at 640 nm, 561 nm and then 488 nm. Exposure time varied depending on sample intensity, but remained the same for each wavelength between the two samples of infected cells (FL-ENaC and ΔENaC), 400 ms for 640 nm, 2 s for 561 nm and 600 ms for 488 nm. Images were imported into Fiji for image analysis.

## Acknowledgements

We thank the Multiscale Microscopy Core (OHSU) and the Advanced Light Microscopy Core (OHSU) for support with microscopy and the Advanced Computing Center (OHSU) for computational support. We are grateful to L Vaskalis for assistance with figures, to D Cawley for monoclonal antibody production, to D Ellison, M Mayer and R Goodman for advice, and to A Bouneff for naming the GRIP domain. We also want to thank the members of the Gouaux and Whorton labs for helpful discussions. This work was supported by the NIH (IB, DP5OD017871).

## Additional information

### Funding

| Funder | Grant reference number | Author |
| --- | --- | --- |
| National Institutes of Health | DP5OD017871 | Isabelle Baconguis |

The funders had no role in study design, data collection and interpretation, or the decision to submit the work for publication.

### Author contributions

Sigrid Noreng, Data curation, Formal analysis, Validation, Investigation, Visualization, Methodology, Writing—original draft, Writing—review and editing, Performed sample preparation, Collected, analyzed, and processed cryo-EM data, Carried out electrophysiology and confocal microscopy experiments; Arpita Bharadwaj, Conceptualization, Formal analysis, Validation, Visualization, Methodology, Writing—original draft, Writing—review and editing, Performed sample preparation and collected cryo-EM data; Richard Posert, Validation, Investigation, Visualization, Methodology, Writing—review and editing, Designed ENaC constructs; Craig Yoshioka, Data curation, Formal analysis, Validation, Investigation, Visualization, Methodology, Writing—original draft, Writing—review and editing, Collected, analyzed, and processed cryo-EM data; Isabelle Baconguis, Conceptualization, Resources, Data curation, Formal analysis, Supervision, Funding acquisition, Validation, Investigation, Visualization, Methodology, Writing—original draft, Project administration, Writing—review and editing,

Carried out preliminary characterization and purification, Performed electrophysiology, model-building and refinement

### Author ORCIDs
Sigrid Noreng http://orcid.org/0000-0001-5767-1399
Arpita Bharadwaj http://orcid.org/0000-0002-3867-7610
Richard Posert http://orcid.org/0000-0001-9010-2104
Craig Yoshioka https://orcid.org/0000-0002-0251-7316
Isabelle Baconguis http://orcid.org/0000-0002-5440-2289

### Decision letter and Author response
Decision letter https://doi.org/10.7554/eLife.39340.042
Author response https://doi.org/10.7554/eLife.39340.043

## Additional files

### Supplementary files
• Transparent reporting form
DOI: https://doi.org/10.7554/eLife.39340.034

### Data availability
The three-dimensional cryo-EM density map and the coordinate for the structure of ΔENAC have been deposited in the EMDataBank and Protein Data Bank under the accession codes EMD-7130 and 6BQN, respectively.

The following datasets were generated:

| Author(s) | Year | Dataset title | Dataset URL | Database, license, and accessibility information |
|---|---|---|---|---|
| Noreng S, Bharadwaj A, Posert R, Yoshioka C, Baconguis I | 2018 | ΔENaC model coordinates | https://www.rcsb.org/structure/6BQN | Publicly available at RCSB Protein Data Bank (accession no. 6BQN) |
| Noreng S, Bharadwaj A, Posert R, Yoshioka C, Baconguis I | 2018 | ΔENaC map, FSC | http://emsearch.rutgers.edu/atlas/7130_summary.html | Publicly available at the Electron Microscopy Data Bank (accession no. EMD-7130) |

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
