## [Decision Letter]

Thank you for submitting your article "Structure of the human epithelial sodium channel by cryo-electron microscopy" for consideration by *eLife*. Your article has been reviewed by three peer reviewers, and the evaluation has been overseen by Sriram Subramaniam as Reviewing Editor and Richard Aldrich as the Senior Editor. The following individual involved in review of your submission has agreed to reveal his identity: Lawrence Palmer (Reviewer #1).

The reviewers have discussed the reviews with one another and the Reviewing Editor has drafted this decision to help you prepare a revised submission.

Summary:

The reviewers are in general agreement that this is an important study that reports the first ENaC structure and provides a framework for future work on this protein. It also addresses a specific topic of interest in the field, namely the location of fragments cleaved from the α and γ subunits during proteolytic activation of the channel. Overall the work is impressive, but there are concerns that important gaps remain on technical and biological fronts.

All three reviewers were concerned about the accuracy of the current model, particularly in the membrane-spanning regions, where no glycosylation sites can be used to guide the chain fitting. However, given the significance of the structure to this field, and the potential value of having structural findings available to biochemists and physiologists working on these channels, there was consensus that the authors should have an opportunity to submit a revised version. The authors should make an attempt to include in the revised version either additional mutagenesis data, or rigorous local comparisons of the model with the density, or perhaps better 3D classification, or at least address in some depth the limitations of the atomic model reported.

Essential revisions:

1) The TMD of the structure is poorly resolved, preventing analysis of the ion selectivity and permeation path in this region. The authors state that "The TMD is not very well ordered, hampering our ability to model the TMD and assign a functional state of the channel". However, in the Abstract the authors concluded that the reported structure was in a resting state. While this conclusion may be inferred from the functional study, this statement should be tempered or removed from the Abstract. On the other hand, the 2D class average result in Figure 1E suggests that the TMD is not disordered. Based on Figure 3——figure supplement 1, the final reconstruction is from about 250k particles, which is large enough for a further 3D classification. Another round of 3D classification with a local mask on the TMD may further improve the resolution of TMD.

2) Disease-related mutations should be discussed. Since gain or loss of function mutations of ENaC are associated with severe diseases, the study will be of more general interest if the authors provide some structure-based mechanistic analysis for at least some of the disease related mutations that can be mapped onto the structure.

3) Does the ΔENaC construct localize to the plasma membrane of the oocytes? It is possible that deletion of the N- and C-termini of all the subunits eliminates important trafficking signals and prevents the protein from reaching the surface. On the other hand, if the protein does get to the surface it suggests that the structure is of a closed state of the channel. This would be nice to know.

4) The use of glycosylation sites to disambiguate the subunits in the cryo-EM map is sensible, and seems well done. However, given the modest resolution of some of these residues in the map, did the authors use an orthogonal approach to disambiguate the subunits? For example, a Western blot against the heterotetramer using 7B1 as the primary antibody would reveal to which subunit it binds, and the same could be done for 10D4.

5) It is evident that the single Fab was essential to break the pseudo-symmetry and make particle alignment possible. However, could the authors clarify why a second Fab was used in the final structure? While two Fabs in combination would have been useful for identifying the subunits, it does not seem that the Fabs were used for this purpose; rather, the glycosylation sites were used.

6) Figure 4—figure supplements 2-5 show very nice high-resolution features in the map. These features are presumably taken from the fully B-factor sharpened maps shown in Figure 3—figure supplement 1, and not from the lower B-factor map presented in Figure 4 and Figure 4—figure supplement 1. To orient the reader, perhaps the authors could show a larger zoomed view of the full, final sharpened map to accompany these high-resolution sub-regions. This would also help give the reader a better impression of the final map.

---

## [Author Response]

Essential revisions:1) The TMD of the structure is poorly resolved, preventing analysis of the ion selectivity and permeation path in this region. The authors state that "The TMD is not very well ordered, hampering our ability to model the TMD and assign a functional state of the channel". However, in the Abstract the authors concluded that the reported structure was in a resting state. While this conclusion may be inferred from the functional study, this statement should be tempered or removed from the Abstract.

We appreciate this point and have revised the sentence in the Abstract to read, “Here we present the structure of human ENaC in the uncleaved state, determined by single-particle cryo-electron microscopy.”

On the other hand, the 2D class average result in Figure 1E suggests that the TMD is not disordered. Based on Figure 3—figure supplement 1, the final reconstruction is from about 250k particles, which is large enough for a further 3D classification. Another round of 3D classification with a local mask on the TMD may further improve the resolution of TMD.

The 2D class averages do not show clear and continuous features representative of membrane-spanning domains (please see a panel from Author response image 1 comparison figure of a 2D class average from a recently published mitochondrial calcium uniporter structure PMID:29954988). Nevertheless, we made great effort to resolve the transmembrane segments, including 3D classification with a local mask on either the entire TMD or the TMD segment near the intracellular side of the membrane-spanning domain. We also performed focused classification of the TMD region and signal subtraction of the ECD. Despite these efforts, we did not obtain improved resolution in the TMD region consistent with unresolvable conformational heterogeneity. We have also revised the text to state “The TMD is not well ordered, hampering our ability to model the entire TMD region and assign a functional state of the channel.” We are committed to resolving the TMD region of ENaC to understand the mechanism of ion selectivity that is vital for its role as the rate-limiting step of Na^+^ reabsorption.

**Author response image 1. respfig1:** Selected 2D class averages of ΔENaC. (**a**) The 2D class averages of ΔENaC do not show clear and continuous features representative of membrane-spanning domains. (**b**) Supplementary Figure 4E from Yoo et al. 2018, PMID:29954988.

2) Disease-related mutations should be discussed. Since gain or loss of function mutations of ENaC are associated with severe diseases, the study will be of more general interest if the authors provide some structure-based mechanistic analysis for at least some of the disease related mutations that can be mapped onto the structure.

We are grateful to the reviewer for pointing this out and have incorporated a brief discussion related to a mutation located in the ECD that underlies Liddle’s syndrome (–subsection “ENaC Structural Overview”, third paragraph). The revised text now states, “Underscoring the importance of the wrist region and the critical roles that disulfide bridges play in maintaining the structural and functional integrity of ENaC, alterations of a conserved cysteine, α-Cys479 to an Arg, causes Liddle Syndrome due to a missense mutation that not only eliminates a disulfide bridge located at the juncture of the thumb and palm domains but also introduces a bulky positively charged residue (Salih et al., 2017) (Figure 5A).” We note that other disease-associated mutations are in regions of the channel that are not resolved or are not present in the structure reported here.

3) Does the ΔENaC construct localize to the plasma membrane of the oocytes? It is possible that deletion of the N- and C-termini of all the subunits eliminates important trafficking signals and prevents the protein from reaching the surface. On the other hand, if the protein does get to the surface it suggests that the structure is of a closed state of the channel. This would be nice to know.

We appreciate this point and have carried out several experiments to define the function and localization of ΔENaC. First, we note that extensive experimentation has found no evidence for ΔENaC ion channel activity in either GnTI^-^ HEK cells or in *Xenopus oocytes*. By contrast, we have measured robust Na^+^-selective and amiloride-sensitive current when FL-ENaC is expressed in GnTI^-^ HEK cells (Figure 2—figure supplement 2A, B). Because HEK cells are better suited to defining whether ΔENaC trafficks to the plasma membrane, we carried out the following experiments on ΔENaC and FL-ENaC expressed in GnTI^-^ HEK cells using confocal microscopy. To ensure robust expression, we transduced the HEK cells with baculovirus encoding the ΔENaC and FL-ENaC proteins, taking advantage of the N-terminal eGFP in the Δγ subunit and the N-terminal eGFP in all three FL subunits to visualize expression, respectively. We used wheat germ agglutinin (WGA) labeled with Alexa Fluor 647 to label the plasma membrane of cells (Figure 2—figure supplement 2C, D). Based on eGFP fluorescence, we observed robust expression of both ΔENaC and FL-ENaC (Figure 2—figure supplement 2C, D). We next employed tetramethylrhodamine (TRITC)-labeled 10D4 mAb, an antibody that we know binds to the extracellular domain of ENaC, to probe the plasma membrane localization of ENaC channels. Indeed, we observed overlapping signals from both eGFP and TRITC-10D4 mAb in cells expressing FL-ENaC but not in cells expressing ΔENaC. Based on the confocal imaging results, we show that ΔENaC is not trafficked to the plasma membrane. We have included these results in Figure 2—figure supplement 2. We have also included text describing the results of the live confocal microscopy in the– first paragraph of the subsection “Functional characterization of ∆ENaC”.

4) The use of glycosylation sites to disambiguate the subunits in the cryo-EM map is sensible, and seems well done. However, given the modest resolution of some of these residues in the map, did the authors use an orthogonal approach to disambiguate the subunits? For example, a Western blot against the heterotetramer using 7B1 as the primary antibody would reveal to which subunit it binds, and the same could be done for 10D4.

We have made multiple, distinct efforts to identify ENaC subunits. We have used 12 sites of glycosylation and 16 Phe, Tyr and Trp residues, in addition to 11 well ordered Arg residues, all of which are subunit specific, to conclusively identify subunits in this trimeric complex. We have provided examples of differences in the palm domain between the three subunits and the corresponding positions of the aromatic and basic residues (Figure 4—figure supplement 1D-F).

In addition, we have used our mAbs to independently identify the subunits. Even though the mAbs 7B1 and 10D4 recognize tertiary epitopes and thus are not robust probes for Western blotting, as described in the last paragraph of the subsection “Design and expression of ΔENaC” in the Results section and in the subsection “Generation and isolation of Fabs” in the Materials and methods section, the 10D4 antibody is weakly Western positive. By Western blot analysis we show that the 10D4 mAb recognizes a band near the 75 kDa MW marker, at a position that we have previously shown corresponds to the β subunit (Author response image 2), thus providing evidence that the 10D4 mAb binds to the β subunit. We next tested whether 7B1 and 10D4 bind to homomeric FLα by FSEC and found that only 7B1 binds to homomeric FLα based on the peak shift (Author response image 2). Together, our Western blot and FSEC results confirm our previous conclusion that 7B1 and 10D4 bind to α and β subunits, respectively. In combination with our analysis of the density maps, we have evidence to unambiguously identify the subunits in the complex.

**Author response image 2. respfig2:** The mAbs 7B1 and 10D4 bind to the α and β subunits, respectively. (**a**) Western blot of purified ∆ENaC probed with the 10D4 mAb. The band detected resides near the 75 kDa MW marker indicating that the mAB 10D4 binds to an epitope in the β subunit. (**b**) Representative traces of FL-α (black) expressed in oocytes. The 7B1 Fab binds to FL-α based on the peak shift to the left (red) while 10D4 Fab does not (blue).

5) It is evident that the single Fab was essential to break the pseudo-symmetry and make particle alignment possible. However, could the authors clarify why a second Fab was used in the final structure? While two Fabs in combination would have been useful for identifying the subunits, it does not seem that the Fabs were used for this purpose; rather, the glycosylation sites were used.

Two Fabs were used because the best particle distributions and ice quality were obtained with the di-Fab complex. We nevertheless appreciate this comment and have revised the text to clarify that the use of two different Fabs versus one Fab was due to differences in grid conditions and the resulting data quality. Most importantly, we obtained superior density maps using ΔENaC in complex with Fabs 7B1 and 10D4. The revised text now states, “We monitored and compared grid conditions and the resulting data quality between the monoFab and the diFab complexes of ENaC which include ice thickness, sample quality, particle distribution, and orientation, and discovered that the diFab complex was a more promising complex for cryo-EM analysis.”

6) Figure 4—figure supplements 2-5 show very nice high-resolution features in the map. These features are presumably taken from the fully B-factor sharpened maps shown in Figure 3—figure supplement 1, and not from the lower B-factor map presented in Figure 4 and Figure 4—figure supplement 1. To orient the reader, perhaps the authors could show a larger zoomed view of the full, final sharpened map to accompany these high-resolution sub-regions. This would also help give the reader a better impression of the final map.

We agree with the reviewer and provided additional figure panels in Figure 4—figure supplement 1D-O as well as a video to help orient the reader.